# Dynamic memory to alleviate catastrophic forgetting in continual learning with medical imaging

Matthias Perkonigg [1], Johannes Hofmanninger[1], Christian J. Herold[1], James A. Brink[2], Oleg Pianykh [2], Helmut Prosch[1] & Georg Langs [1]✉

Medical imaging is a central part of clinical diagnosis and treatment guidance. Machine learning has increasingly gained relevance because it captures features of disease and treatment response that are relevant for therapeutic decision-making. In clinical practice, the continuous progress of image acquisition technology or diagnostic procedures, the diversity of scanners, and evolving imaging protocols hamper the utility of machine learning, as prediction accuracy on new data deteriorates, or models become outdated due to these domain shifts. We propose a continual learning approach to deal with such domain shifts occurring at unknown time points. We adapt models to emerging variations in a continuous data stream while counteracting catastrophic forgetting. A dynamic memory enables rehearsal on a subset of diverse training data to mitigate forgetting while enabling models to expand to new domains. The technique balances memory by detecting pseudo-domains, representing different style clusters within the data stream. Evaluation of two different tasks, cardiac segmentation in magnetic resonance imaging and lung nodule detection in computed tomography, demonstrate a consistent advantage of the method.

[1] Department of Biomedical Imaging and Image-guided Therapy, Medical University of Vienna, Vienna, Austria. [2] Department of Radiology, Massachusetts General Hospital, Harvard Medical School, Boston, MA, USA. ✉email: georg.langs@meduniwien.ac.at

Deep learning (DL) algorithms are rapidly gaining relevance in medical imaging, enabling computational segmentation[1,2], classification or detection[3] of anatomical structures and anomalies[4] relevant for diagnosis, prediction, or prognosis. In some cases, their capabilities surpass even those of human experts[5,6], making them a central tool in the advancement of using imaging data for diagnosis, and for supporting treatment decisions.

However, clinical imaging technology, diagnostic workflows, and even imaging markers of diseases are not static. Instead, they are subject to an ever-evolving environment in which DL algorithms have to adapt in order to remain relevant. Currently, DL models are trained once, yielding an impressive performance on images comparable to their training experience. Yet, their sustainability is limited, as they become outdated while technology advances[7]. This inability to adapt to new data that is different from the training data in some aspect (dataset shifts) severely hampers their utility and adoption in clinical practice.

Dataset shifts occur when the training data distribution differs from the distribution of data at model inference[8,9]. One type of such shift, domain shift (or acquisition shift) might occur due to technical progress in scanner technology. In clinical practice, and consequently, in studies involving medical imaging data, acquired data frequently originate from different scanners, scanner generations, manufacturers, or imaging protocols. To successfully adapt deployed deep learning models to the changing environment, it is crucial to develop and advance methods that consider these domain shifts.

Here, we focus on accounting for domain shifts occurring at unknown times in a continuous data stream, reflecting clinical practice. A DL model is trained on a set of images acquired by a single scanner (base training) and subsequently updated continuously to changes in image appearance that occur in a data stream as new scanners are added. At the same time, knowledge about previously seen domains should not be forgotten as new domain information is incorporated in the model. Figure 1 illustrates the general setting of this work. The model is trained to convergence on a base training set of domain A data; afterwards, it is exposed to a continuous data stream in which, after some time, domain B, C and D data appear. Without updating the model after base training (static deep learning), accuracy on later domains suffers, since they leave the distribution of the training data. See below for an example, where the base model failed to segment images from subsequent scanners. Continual learning methods counter this effect.

The focus of continual learning (also referred to as lifelong learning) are machine learning techniques for accumulating the ability to handle new tasks (or, in the context of this work, new domains) in a model[10,11]. A major undesired effect counteracted by continual learning methods is catastrophic forgetting, when updating a model to learn a new task would lead to a deterioration of performance on previous tasks[12]. Ideally, continual learning could yield improvements of performance on previous tasks when training on subsequent tasks, a desirable effect known as positive backward transfer resulting from the increased variety of training examples the model is exposed to[13].

We propose dynamic memory (DM) as a continual learning method, to deal with the emergence of new data sources at unknown time points in a continuous stream of medical images (Fig. 1). DM is a rehearsal method, which keeps a small, diverse subset of the data stream in memory to alleviate catastrophic forgetting. DM diversifies the memory using a style metric to maintain images with a variety of styles observed in the continuous data stream. As an optional module, we utilize a pseudo-domain (PD) model to detect clusters of a similar style from the continuous stream. Those pseudo-domains can be seen as proxies for the unknown, real domains and are used to balance the memory and training process (DM-PD). To demonstrate the generalizability of our method, we apply it to two different tasks with different imaging modalities. First, we perform cardiac segmentation in magnetic resonance imaging (MRI), and second, we apply our approach to lung nodule detection in computed tomography (CT). We show that on both tasks, our method outperforms continual learning baseline methods. Note that we are not focusing on the development of a new single scanner state-of-the-art method for either of the tasks, but rather we want to show how a continual learning method can be applied to adapt a model to a continuous stream of imaging data, including domain shifts, without explicit domain knowledge.

## Results

### Data sets

*Cardiac segmentation.* Experiments were performed on data from a multi-centre, multi-vendor challenge data set[14]. The data set included data from four different vendors Siemens, General Electric, Philips, and Canon. We considered each of those vendors as one domain. We split the data into Base training, Continual training, validation and test set on a patient level. Table 1a shows the number of individual slices for each domain in those data set splits.

*Lung nodule detection.* For lung nodule detection we used data extracted from the LIDC-database[15], with the annotations as provided for the LUNA16-challenge[16]. In addition, we used the LNDb challenge data set[17]. For all lung nodule annotations, we constructed bounding boxes around the annotated lesion and extracted 2D slices with lesions. To demonstrate our continual learning with shifting domains, we constructed a data set of the three most common domains, in terms of scanner vendor and reconstruction kernel, in LIDC and as a fourth domain, the LNDb data set. For LIDC, the most commonly used settings including lesions were GE Medical Systems with low-frequency reconstruction algorithm (GE/L, $n = 527$), GE Medical Systems with high-frequency reconstruction algorithm (GE/H, $n = 215$) and Siemens with B30f kernel (Siemens, $n = 130$). The LNDb data set used multiple Siemens scanners. To match the nodule definition in the LIDC database we excluded nodules with a diameter < 3 mm, resulting in a total of 625 images. Those images were split into base training, continual training, validation and test data set according to Table 1b analogous to the cardiac segmentation experiment.

### Dynamic memory alleviates catastrophic forgetting for cardiac segmentation

To evaluate the ability of dynamic memory to achieve good performance while counteracting catastrophic forgetting, we performed cardiac segmentation on 2D MRI slices as multi-label segmentation with three labels: Left ventricle (LV), right ventricle (RV) and left ventricular myocardium (MYO). Images were acquired with scanners of four different vendors, in order of their appearance in the data stream: Siemens (A), GE (B), Philips (C), and Canon (D). We refer to them as Scanner A–D to facilitate understanding of the order. Base training was done on Scanner A data only; subsequently, the model was trained on a continuous data stream in which the image domains gradually changed from Scanner A to D (Fig. 1). We compared different continual learning strategies: (1) the DM method, (2) the DM method with pseudo-domain detection (DM-PD), (3) a random memory replacement strategy, in which every new sample replaced a randomly chosen sample currently in memory (Random) and (4) a naive approach of learning on a data stream without counteracting catastrophic forgetting (Naive). In

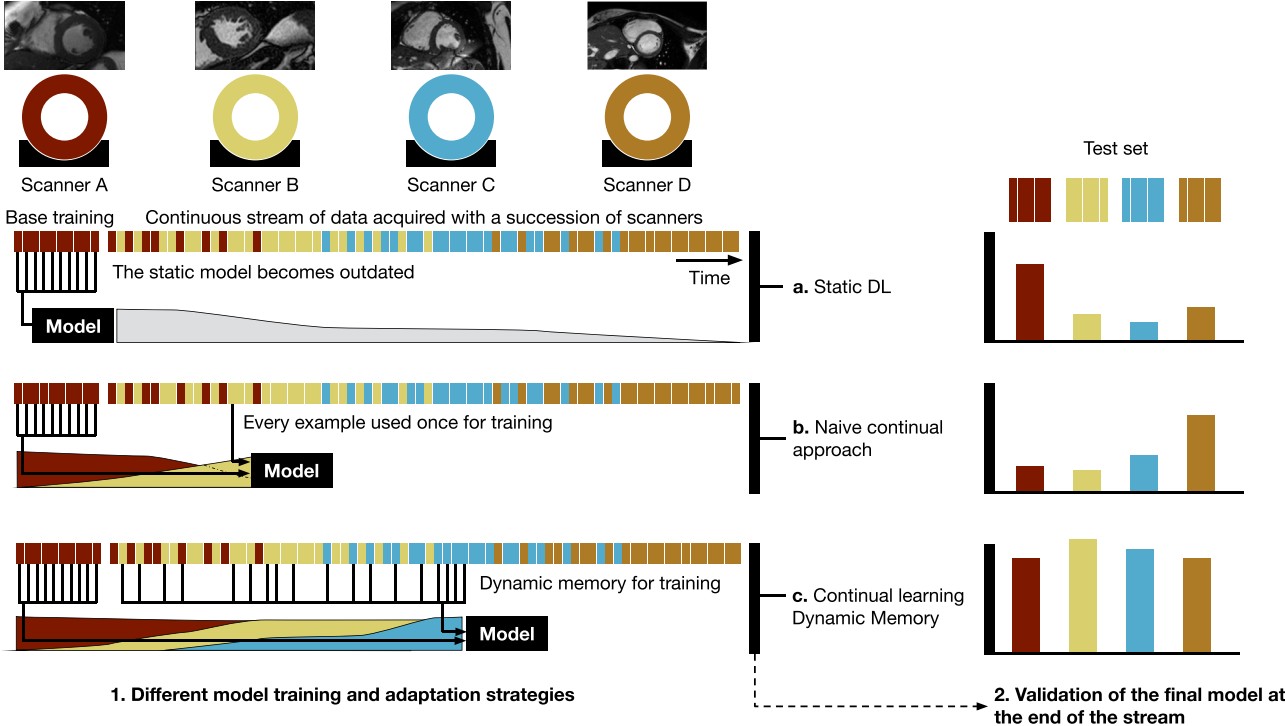

**Fig. 1 Different model training and adaptation strategies. 1** Continual DL for image analysis adapts to new data properties and, at the same time, retains the capability to work with older data. **a** Static DL: after training and deploying a DL model, technology changes and the accuracy of the model decreases. **b** A naive continual learning approach to solve this limitation is to continue training a model on a stream of data. However, this leads to forgetting of old data properties, and a corresponding decrease in performance on these data. **c** As a continual learning approach, dynamic memory recognizes new emerging domains, and samples data in a continuous stream accordingly. The ML model adapts to new technology but stays accurate on the diverse set of scanners previously seen. This is important to ensure backward compatibility of the model and to build a more stable model that adapts to new scanners faster. **2** Validation on a separate test set evaluates the DL model performance at the end of the period.

**Table 1 Number of slices in data sets for both experimental tasks.**

**(a) Cardiac segmentation data set**

|  | Siemens | GE | Philips | Canon |
|---|---|---|---|---|
| Base | 1120 | 0 | 0 | 0 |
| Continual | 614 | 720 | 2206 | 758 |
| Validation | 234 | 248 | 220 | 258 |
| Test | 228 | 246 | 216 | 252 |

**(b) Lung nodule detection data set**

|  | GE/L | GE/H | Siemens | LNDb |
|---|---|---|---|---|
| Base | 253 | 0 | 0 | 0 |
| Continual | 136 | 166 | 102 | 479 |
| Validation | 53 | 23 | 10 | 55 |
| Test | 85 | 26 | 18 | 91 |

addition, we compared results with state-of-the-art continual learning methods: (5) Elastic Weight Consolidation (EWC)[18], and two methods that require domain knowledge (6) Gradient Episodic Memory (GEM)[19] and (7) experience replay with Maximally Inferred Retrieval (ER-MIR)[20]. Note that DM and DM-PD operate without domain knowledge, representing a more realistic assumption in clinical practice. For comparison, we also trained two baseline models: First, a joint model (JModel) with all training data that treated the entire data set as a single hypothetical static data set; and second, domain-specific models

(DSM) trained in a static training scheme on each of the domains, separately. Finally, we report results for a static base model (Base) trained on Scanner A, and applied to scanners A to D. All methods used the same backbone convolutional neural network (CNN) for segmentation, an FC-ResNet50[21].

We compared the segmentation accuracy of all approaches on a separate test set that contained data from all four scanners after training had finished. Furthermore, we specifically assessed whether model accuracy on one scanner benefits from model training on other scanners by evaluating backward transfer (BWT) and forward transfer (FWT)[19].

In Table 2, the approaches are compared in terms of average Dice score over LV, RV, and MYO for a memory size of $M = 128$. An evaluation of LV, RV, and MYO segmentation separately showed similar trends (see Supplementary Tables 1–3). DM and DM-PD performed similarly and outperformed all other continual learning strategies for which no information about domain membership is required (Naive, Random and EWC). For images of the last domain (Scanner D), EWC had the highest mean Dice score ($0.850 \pm 0.003$) but at the cost of a high negative BWT value ($-0.014 \pm 0.007$), showing that catastrophic forgetting occurred. DM and DM-PD exhibited no forgetting, as indicated by the neutral BWTs of 0.000 and 0.003, respectively. GEM and ER-MIR showed a similar performance than DM-PD, but needed information about domain membership of the individual samples, which is not feasible in clinical practice. Comparing a memory with a random replacement strategy to DM and DM-PD showed that the style metric used for DM was effective to choose samples to form a diverse memory. Random replacement resulted in forgetting of previous domains during the

**Table 2 Cardiac MR segmentation results after continual training measured as an average Dice score (DSC) over LV, RV and MYO segmentation computed on the test set.**

| Meth. | M | Scanner A | Scanner B | Scanner C | Scanner D | BWT | FWT |
|---|---|---|---|---|---|---|---|
| DM (Ours) | 128 | 0.802 ± 0.005 | 0.762 ± 0.002 | 0.807 ± 0.004 | 0.840 ± 0.009 | 0.000 ± 0.002 | 0.032 ± 0.004 |
| DM-PD (Ours) | 128 | 0.799 ± 0.010 | 0.763 ± 0.004 | 0.809 ± 0.005 | 0.844 ± 0.010 | 0.003 ± 0.004 | 0.031 ± 0.005 |
| Random | 128 | 0.786 ± 0.015 | 0.746 ± 0.008 | 0.797 ± 0.005 | 0.847 ± 0.005 | −0.011 ± 0.007 | 0.033 ± 0.004 |
| EWC[18] | | 0.786 ± 0.008 | 0.738 ± 0.014 | 0.792 ± 0.007 | 0.850 ± 0.003 | −0.014 ± 0.007 | 0.032 ± 0.003 |
| Naive | | 0.781 ± 0.013 | 0.726 ± 0.026 | 0.789 ± 0.011 | 0.848 ± 0.003 | −0.018 ± 0.123 | 0.032 ± 0.002 |
| GEM[19] | 128 | 0.798 ± 0.005 | 0.761 ± 0.008 | 0.804 ± 0.003 | 0.846 ± 0.002 | −0.005 ± 0.004 | 0.033 ± 0.003 |
| ER-MIR[20] | 128 | 0.798 ± 0.005 | 0.763 ± 0.007 | 0.808 ± 0.002 | 0.847 ± 0.001 | −0.004 ± 0.003 | 0.036 ± 0.003 |
| DSM | | 0.802 ± 0.017 | 0.748 ± 0.012 | 0.806 ± 0.014 | 0.835 ± 0.005 | – | – |
| JModel | | 0.822 ± 0.010 | 0.798 ± 0.016 | 0.823 ± 0.006 | 0.852 ± 0.007 | – | – |
| Base | | 0.797 | 0.763 | 0.792 | 0.763 | – | – |

± indicates the interval over $n = 5$ independent runs with different seeds. Dynamic memory (DM) is compared to DM with a pseudo-domain module (DM-PD), random replacement strategy (Random), elastic weight consolidation (EWC) and naive continual learning (Naive). Methods requiring domain membership knowledge are gradient episodic memory (GEM), and experience replay with maximally inferred retrieval (ER-MIR). Domain-specific models (DSM), a joint model (JModel) and using base training only (Base) serve as a reference. For base training only one model was trained to avoid the influence of base training results on subsequent continual training, therefore no standard deviations are indicated. For a visual presentation of the results, see Supplementary Fig. 1.

course of continual training (BWT of −0.011), while DM and DM-PD kept a good performance on all scanners.

On data from Scanner B, for which a relatively small sample size of 720 images was used for training, DM and DM-PD were able to achieve good performance without domain membership information. Learning with random replacement, EWC and naive training resulted in a significant performance drop for Scanner B. This demonstrated that by using a Gram matrix-based style metric, DM was less sensitive to the amount of samples per scanner vendor than other continual learning strategies.

JModel was the upper bound of what the training can achieve, especially for domains with few samples, i.e., for Scanner B, the performance gap between continual learning and hypothetical static batch training accessing all data was high (0.763 ± 0.004 vs. 0.798 ± 0.016). This was due to the fact that in continual learning, underrepresented samples were less often seen than in static training. In Supplementary Table 4, different memory sizes of $M = \langle 64, 128, 256, 512, 1024 \rangle$ for DM and DM-PD were compared. Adding more memory resulted in better performance but due to differences in the training dynamics (JModel trains batch-wise and thus can see more batches than in continual learning settings), the performance of JModel was not achievable. DM with a smaller memory of $M = 64$ was not able to accurately capture the diversity of the training data distribution and thus resulted in forgetting (BWT $= -0.005$).

Figure 2 shows how the mean DSC changed in the course of continual training. For EWC and naive continual learning, catastrophic forgetting was observed, and accuracy on previous domains declined as new domains entered model training. For DM and DM-PD, accuracy was more stable across all domains as continual training proceeded.

Qualitative assessment of semantic segmentation in Fig. 3 showed comparable performance results for the different continual learning approaches. However, while naive continual learning showed better results for Scanner D, DM and DM-PD performed well on all scanners including previously seen domains. The comparison with the base model (trained on Scanner A data only) showed that we need a continual learning method to adapt the model to changing visual appearance. The base model failed to perform accurate segmentation for scanners B–D, resulting in a high number of false negatives.

**Dynamic memory alleviates catastrophic forgetting for lung nodule detection.** Lung nodule detection was performed as bounding box detection on 2D CT slices and measured in average precision (AP), as defined in the 'Evaluation' section. Four image domains were included in the data stream: GE with a low-frequency reconstruction algorithm (GE/L); GE with a high-frequency reconstruction algorithm (GE/H); Siemens; and LNDb (see the 'Data sets' section for details), in the following Scanners E–H. Base training was performed on data from Scanner E. Due to the definition of EWC, a direct comparison was not possible for detection tasks. DM and DM-PD were compared to a random replacement memory and a naive continual learning approach. Analogously to segmentation, two state-of-the-art methods requiring domain labels (which is not required by DM and DM-PD), GEM[19] and ER-MIR[20] were evaluated as reference. As baseline models, a joint model (JModel) and domain-specific models (DSM) were compared, and the results for the static base model (Base) were evaluated analogously to the segmentation experiment. As a task network, Faster R-CNN with a ResNet-50 backbone was used[22].

Overall, DM-PD and DM performed better in terms of AP than the naive approach, as seen in Table 3. Both outperformed the naive method and effectively counteracted catastrophic forgetting. For all domains extracted from LIDC (Scanners E, F and G), DM performed well. However, we observed a drop in performance for Scanner H for all methods. This drop was caused by a population shift in addition to the large domain shift. In LIDC data, lung nodules had a mean diameter of 8.29 mm, while Scanner H data (extracted from LNDb) included smaller lesions with a mean diameter of 5.99 mm. By design, DM does not detect population shifts (i.e., the change of lesion characteristics as opposed to imaging characteristics), and thus, could not adapt to Scanner H data quickly. The random replacement strategy struggled with learning domains that were present less frequently in the training (Scanner F and G) as they were replaced over time by Scanner H data, resulting in forgetting on those scanners. This effect was less severe for Scanner E data since the base training was performed on data of this scanner. DM and DM-PD counteracted this forgetting by using a style-based metric to diversify the memory and thus kept samples of all scanners in memory. DM-PD performed better than DM without pseudo-domain detection, demonstrating that balancing the training process was an important step for our continual learning method. DM-PD showed the best performance in terms of AP and outperformed the naive approach by around 0.05 AP for Scanners E, F and G. Furthermore, the best backward and forward transferability was observed for DM-PD/128. Thus, it was the preferable model for lung nodule detection.

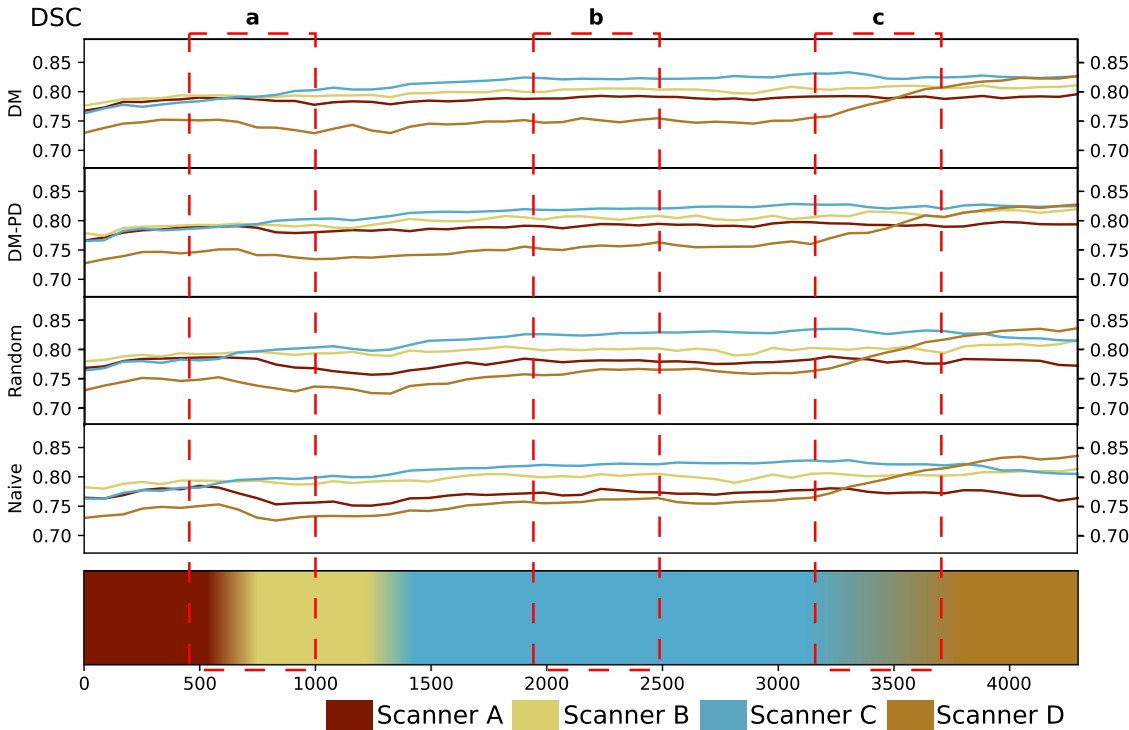

**Fig. 2 Cardiac MR segmentation.** Dice score (DSC) on the validation set during training for $M = 128$ for DM and DM-PD, compared to random replacement and naive continual learning. The timeline at the bottom represents a continuous data stream and the change of domains in the stream. **a** A drop of DSC for Scanner A can be observed when Scanner B occurs in the stream, DM and DM-PD were able to recover from this drop using the memory; **b** all methods are stable during training with Scanner C, and **c** as soon as Scanner D data flows in, we see a quick rise of DSC for Scanner D validation samples. Naive and random replacement lose some DSC points during that period, while DM remains stable. This significant and rapid change shows that Scanner D is different compared to the others and not as closely related as Scanner B and C.

Supplementary Table 5 shows results of comparing different memory sizes $M = \langle 64, 128, 256, 512, 883 \rangle$ (883 corresponds to storing all samples of the continuous stream to memory). For DM, a larger memory size was beneficial compared to a smaller memory size. For DM-PD, the results showed that the smaller the memory ($M = 64$ and $M = 128$) the more performance gain was achieved by pseudo-domain (PD) detection compared to mere DM. For larger memory sizes, the benefit of PDs vanished. Due to differences in the training sequence, the performance DM with $M = 883$ was different from those of JModel. While DM/883 performed better on Scanner E and F, JModel showed higher AP values for Scanner G and H.

In Fig. 4, the change in validation performance during training is depicted for DM and DM-PD with $M = 128$ compared to random replacement memory and naive continual learning. While the random replacement and naive strategy showed forgetting, especially for Scanner G, DM and DM-PD kept the performance high without catastrophic forgetting.

To analyse the performance of DM in lung nodule detection in detail, precision–recall curves for DM and DM-PD with $M = 128$ were compared to naive continual learning and a base model trained on data from the first scanner E only (Fig. 5a). The base model performed worse than continual learning approaches for all domains, even on test data from the domain which the model was trained on. This showed that knowledge from subsequent scanners can improve final model performance on Scanner E. As expected, the base model's performance deteriorated for the subsequent scanners. The precision–recall curves of the naive continual learning approach showed improvement over the base model. Compared to DM and DM-PD, it exhibited a worse performance for scanners E–G and a slightly better performance for scanner H. This illustrates how naive continual

learning could adapt to new scanners but—in contrast to DM and DM-PD—suffered from forgetting, while updating the model to scanner H data.

In Fig. 5b bounding box detections for all four domains are shown. Overall, a higher number of false positives occurred for the naive approach compared to DM and DM-PD. Given the fact that we performed detection on 2D slices only, DM and DM-PD showed a good overall performance. For lung nodule detection, we showed a clear benefit using rehearsal with our DM method in continual learning settings with unknown domain shifts.

**Pseudo-domain detection maintains a more balanced memory.** For lung nodule detection, we analysed differences between training DM with the pseudo-domain module versus training without the pseudo-domain module for $M = 128$. First, we evaluated how the samples in memory at the end of training were distributed compared to the whole training corpus by embedding the Gram matrices of all training samples to an embedding space using t-distributed stochastic neighbour embedding (TSNE)[23]. Figure 6a shows a clear distinction between the domains of Scanner F, Scanner H, and Scanners E and G. Scanners E and G were close according to their style due to the similar reconstruction kernel used for those domains. The markers in the figure indicate the samples in memory at the end of continual training. For DM-PD, those were more equally distributed over the whole training set.

This observation was confirmed by data depicted in Fig. 6c where we observed a clear over-representation of the first domain (Scanner E) over all subsequent domains for training with DM only, compared to balancing with pseudo-domains (DM-PD).

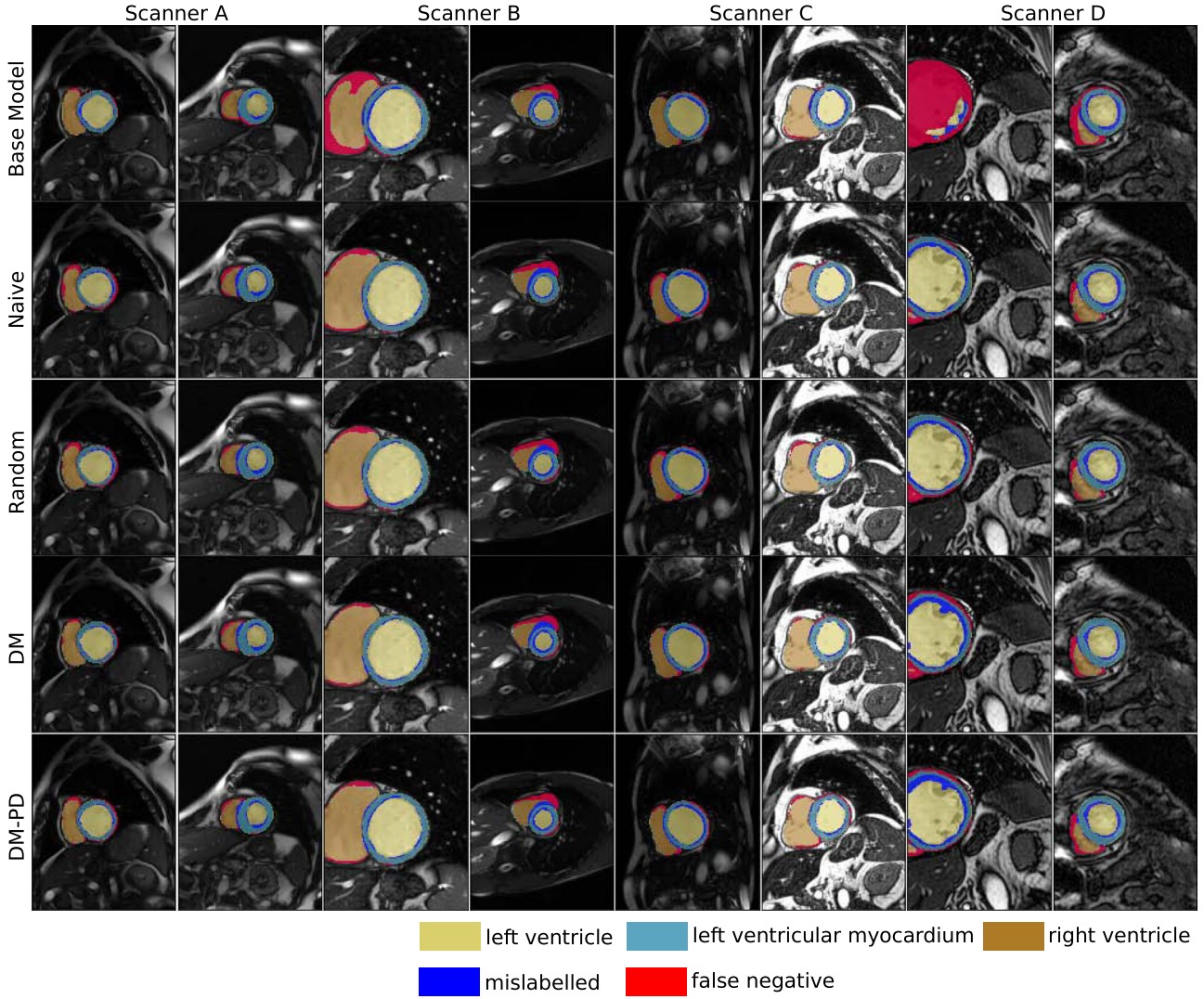

**Fig. 3 Qualitative examples for cardiac segmentation.** Results for DM and DM-PD with $M = 128$ compared to naive continual learning and random memory replacement. Mislabelled refers to pixels that were labelled, but the class membership was confused by the model. The base model was trained in a static training approach on data from Scanner A only.

**Table 3 CT Lung nodule detection results after continual training measured in average precision (AP) computed on the test set.**

| Meth. | M | Scanner E | Scanner F | Scanner G | Scanner H | BWT | FWT |
|---|---|---|---|---|---|---|---|
| DM (Ours) | 128 | 0.722 ± 0.020 | 0.526 ± 0.021 | 0.592 ± 0.041 | 0.330 ± 0.015 | 0.030 ± 0.018 | 0.063 ± 0.016 |
| DM-PD (Ours) | 128 | 0.750 ± 0.006 | 0.565 ± 0.067 | 0.624 ± 0.024 | 0.355 ± 0.038 | 0.028 ± 0.019 | 0.066 ± 0.030 |
| Random | 128 | 0.752 ± 0.019 | 0.514 ± 0.021 | 0.600 ± 0.021 | 0.394 ± 0.013 | 0.007 ± 0.016 | 0.084 ± 0.026 |
| Naive | | 0.682 ± 0.014 | 0.506 ± 0.017 | 0.561 ± 0.020 | 0.369 ± 0.008 | 0.000 ± 0.008 | 0.091 ± 0.027 |
| GEM[19] | 128 | 0.754 ± 0.012 | 0.568 ± 0.022 | 0.622 ± 0.038 | 0.366 ± 0.024 | 0.034 ± 0.016 | 0.067 ± 0.018 |
| ER-MIR[20] | 128 | 0.754 ± 0.012 | 0.588 ± 0.038 | 0.611 ± 0.039 | 0.363 ± 0.027 | 0.031 ± 0.016 | 0.075 ± 0.016 |
| DSM | | 0.653 ± 0.047 | 0.441 ± 0.074 | 0.643 ± 0.067 | 0.454 ± 0.096 | – | – |
| JModel | | 0.716 ± 0.063 | 0.522 ± 0.114 | 0.711 ± 0.058 | 0.419 ± 0.087 | – | – |
| Base | | 0.645 | 0.372 | 0.509 | 0.136 | – | – |

± indicates the interval over $n = 5$ independent runs with different seeds. Dynamic memory (DM) is compared to DM with a pseudo-domain module (DM-PD), naive continual learning, random replacement strategy (Random), domain-specific models (DSM), a joint model (JModel) and using base training only (Base). In addition, GEM and ER-MIR are shown for reference, noting that they require information about domain membership. For base training only one model was trained to avoid influence of base training results on subsequent continual training, therefore no standard deviations are indicated. For a visual presentation of the results, see Supplementary Fig. 2.

Figure 6b shows an analysis of the amount of domain elements assigned to pseudo-domains. For the example of a single training run, five pseudo-domains were detected. PD-1 represented the real domain of Scanner E. PD-4 and PD-5 represented samples from two scanners, F and G. In PD-2 and PD-3, no clear distinction of domains were represented. Overall, we observed

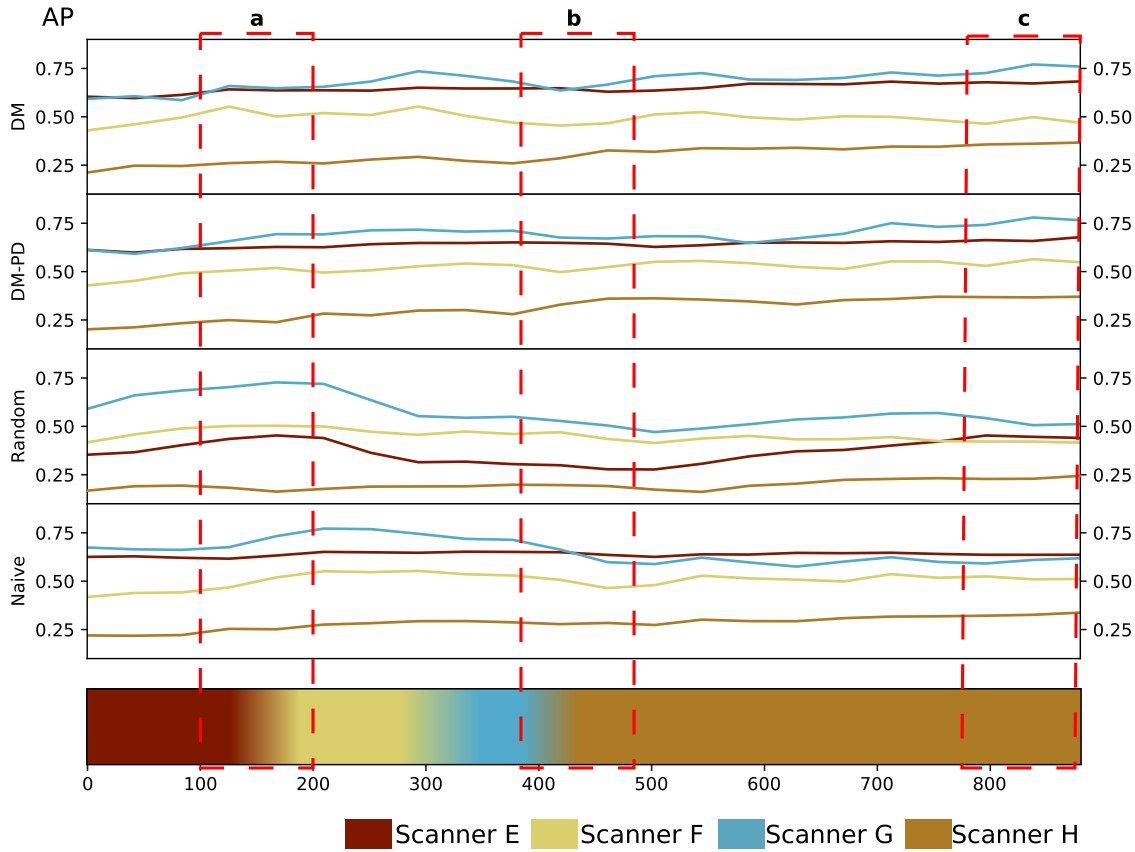

**Fig. 4 Lung nodule detection.** Average precision (AP) measured on the validation set during training for DM and DM-PD with $M = 128$ as well as a random replacement memory and naive continual learning approach. The timeline at the bottom represents the changes of domains in the data stream. **a** As soon as Scanner F data occurred in the stream, the validation performed on the Scanner F (and also Scanner G) domain increased for all approaches. **b** A clear drop in performance (AP) occurred for the naive and random replacement approach for Scanner E, F and G, after some steps of training on Scanner H data, which marked catastrophic forgetting. For DM, the performance first dropped slightly, but recovered after some training steps, because samples from the memory are used for training. The DM-PD performance remained stable for Scanners E, F and G. **c** At the end of continual training, a better performance was achieved for Scanner E, F and G, when dynamic memory was utilized. For Scanner H, the performance for all three approaches was similar. **d** The performance for Scanner E was stable during the entire continual training process for all approaches, showing a base training that was saturated for scanner E.

that using the pseudo-domain module was beneficial to maintain a balanced memory, which was representative of the whole training set distribution.

## Discussion

Machine learning is expanding the use of medical imaging data for diagnosis and prognosis. Advances in deep learning enable the computational detection, segmentation, and classification of entities associated with disease, thus informing individual treatment decisions. After the first iteration of static DL, models has been proven to be effective, the challenge is now to make them sustainable in an environment of continuous advances in image acquisition technology, protocols, or even treatment options. Here, we show that an approach that maintains a diverse dynamic memory could adapt models to changing imaging technology, as it coped with domain shifts. Importantly, while the model learned from new data, it retained the diversity of a rehearsal memory, to remain accurate and reliable across the entire repertoire of imaging sources it had seen. Furthermore, we observed that model knowledge was successfully transferred across scanners. Including training data from other scanners yielded benefits for model accuracy on an individual scanner.

Domain shifts due to scanner variability and their detrimental effects on machine learning (ML) algorithms have been observed

in different image modalities, such as computed tomography (CT) and magnetic resonance imaging (MRI). For CT, the influence of scanners and reconstruction parameters on ML predictions as well as on human annotations, have been studied for chest CT examinations. Demonstrating that scanner variability has a negative influence on radiomics[24,25] and other imaging features[26], this needs to be considered when designing ML models. In ref. [27], the effect of using multiple MRI scanners on ML algorithms for brain MRI was empirically evaluated. To reduce the effect of scanner variability in longitudinal, multi-scanner MRI studies, harmonization[28,29] has been applied. However, different from our work, those methods assume that all data is available at once, which is not the case for a model deployed in clinical practice.

Previously, various methods have been proposed to alleviate catastrophic forgetting[10,11] in continual learning settings. These approaches can be divided broadly into three categories: rehearsal and pseudo-rehearsal methods[19,30–33], regularization-based approaches[18,34,35] and parameter isolation methods[36,37]. For a detailed review, see refs. [10,11]. The majority of those approaches are incremental task learning methods. They focus on learning new tasks incrementally without forgetting the knowledge required for previous tasks. Lately, methods that have focused on accounting for domain shifts have been proposed[38–40]. Domain adaptation (DA) is a related area of research dealing with domain shifts[41–43]. DA focuses on

**a**

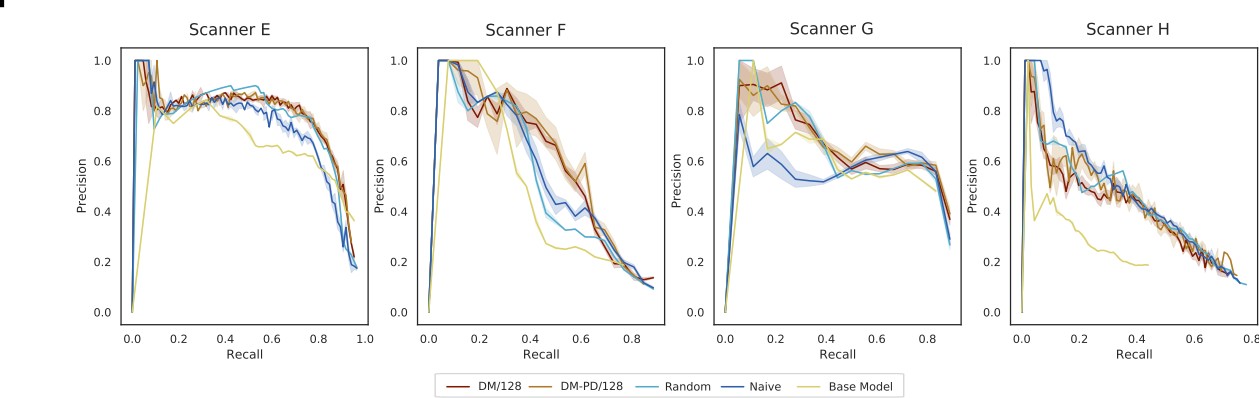

**b**

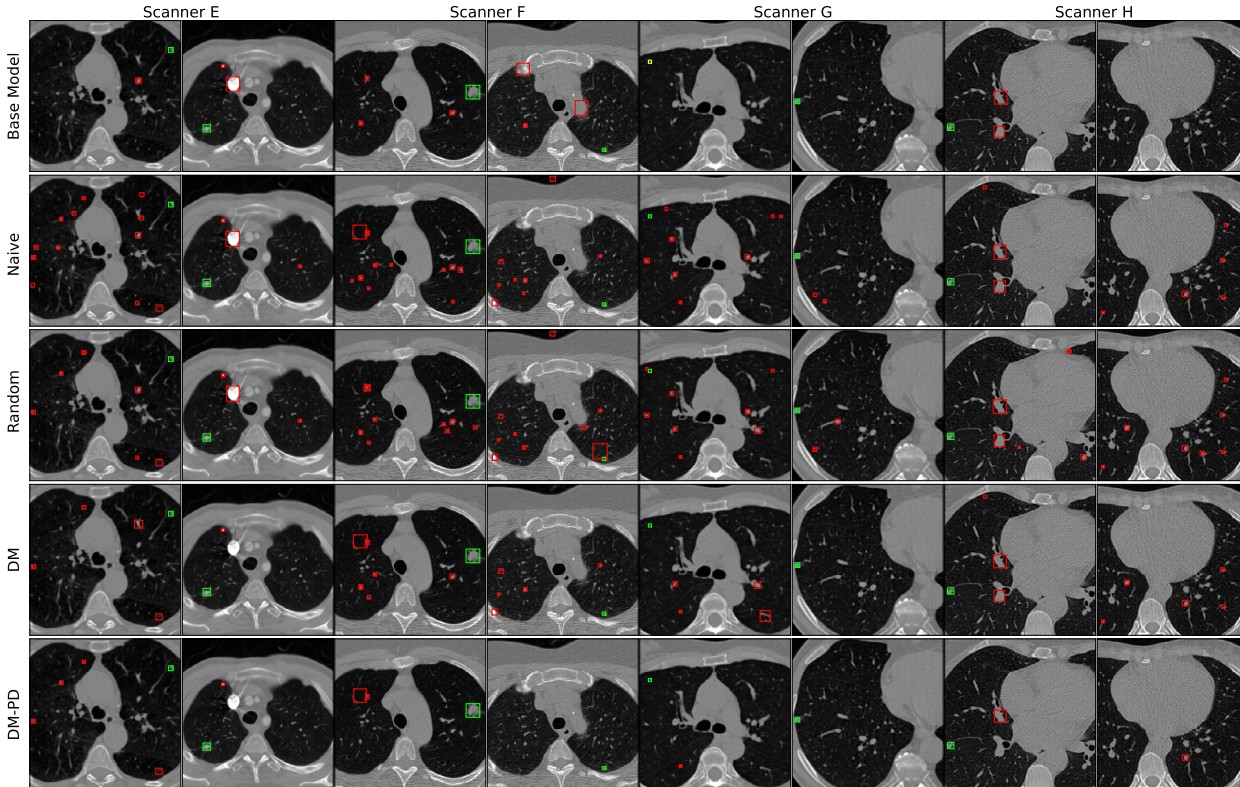

**Fig. 5 Quantitative and qualitative results for lung nodule detection. a** Precision–recall curves of the final naive, DM and DM-PD trained model compared to a base model trained on only scanner E data. Shaded areas represent confidence intervals for $n = 5$ independent training runs. **b** Samples of lung nodule detection of the final naive, DM and DM-PD trained model compared to a base model trained on only scanner E data on all four domains. Green boxes indicate true-positives, yellow boxes false-negatives and red boxes false-positives.

adapting the knowledge learned on one or multiple source domains to a target domain. In medical imaging, DA is used to adapt between different imaging modalities or different image acquisition settings[44]. It assumes access to the source and target domain at once, which is not applicable to a setting using a continuous data stream. Furthermore, DA require the knowledge of a domain- or task membership for each sample. This assumption is not realistic in real-world medical imaging. There the variability of metadata encoding image acquisition information, does not directly map to comparability of imaging characteristics[13]. Since we do not assume to have access to this knowledge, these methods are not applicable for continual learning in the clinical routine. Hence, to date, they have not been adopted in practice. A third related area is transfer learning[45]. Here, an existing model is transferred to a new task or domain by fine-tuning on new data. The sole aim is a model performing well there,

regardless of its capability to work well on the initial domain. During fine-tuning no data from the initial domain is necessary.

Our results show that the capacity of static models to segment and detect is limited when data is acquired with image acquisition machines outside the initial training distribution. At the same time, naive approaches that continuously train on new scanners forget old imaging characteristics, losing their ability to process data with previous acquisition characteristics. By continuously including new training data, while maintaining a diverse rehearsal set, dynamic memory yields good performance across the entire set of observed scanners. The detection of pseudo-domains, representing sub-cohorts that exhibit similar style or imaging characteristics, yields groups of images that correspond to scanners, or groups of scanners that share similar appearance properties. Their detection and injection into the training process for

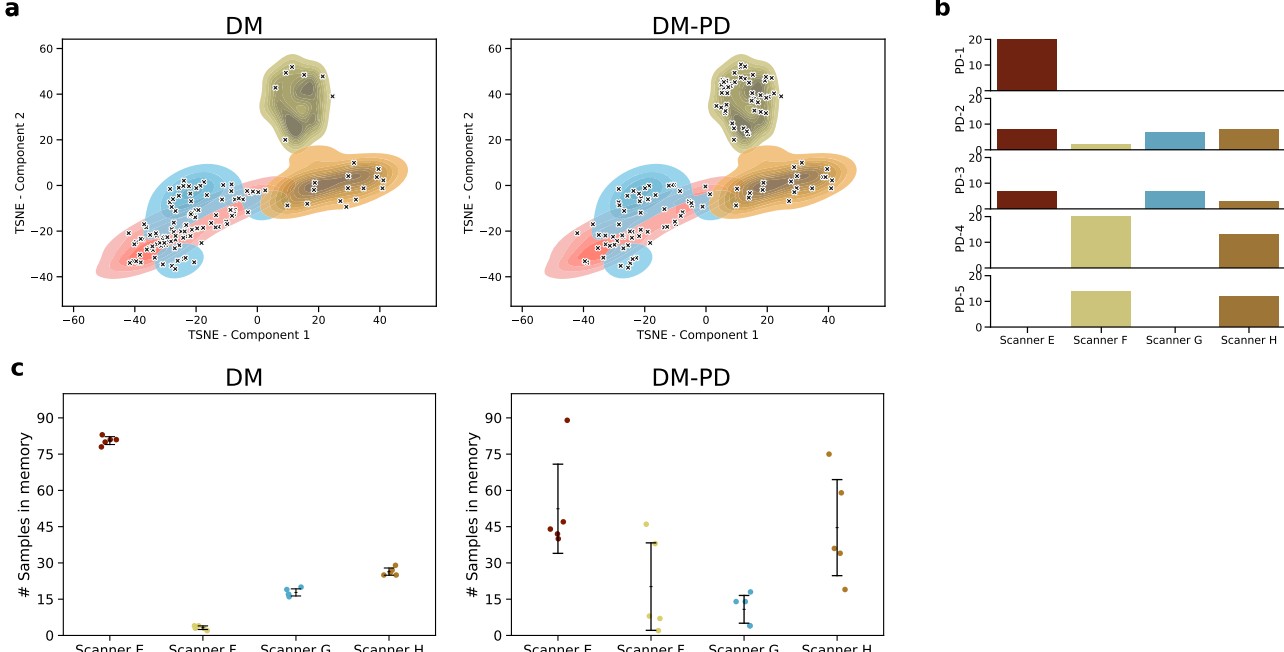

**Fig. 6 Enhancing memory diversity.** Comparison of memory composition for DM vs. DM-PD for $M = 128$. **a** TSNE over the Gram matrices of the training set shows a distinction of the domains. Markers show positions of memory elements at the end of training for one run. DM-PD shows a more equal distribution over the whole training set. **b** Five pseudo-domains detected were detected for a single training run of DM-PD; bars show the amount of domain elements assigned to the pseudo-domains. **c** Amount of domain elements in memory after training for DM and DM-PD. Error bars represent standard deviation and mean (middle line), $n = 5$ independent runs.

DM further improve model performance. Importantly, these pseudo-domains can span multiple scanners if they share imaging characteristics. The usage of a style-based metric alleviates the need for domain membership knowledge, and demonstrates similar performance as state-of-the-art continual learning methods that use this information.

A further benefit of the approach compared to the training of scanner-specific models, is that model performance on a particular scanner typically benefits from training on other scanners. Feature representations of the model obtain better generalizability from training on related but different data. This leads to the processing of new scanner data that profits from training on older scanners (forward transfer), and vice versa (backward transfer). Results were consistent across different imaging modalities (MRI, CT) and image analysis tasks (segmentation, detection).

Results from our cardiac MRI segmentation proved that DM is beneficial compared to naive continual learning, exhibiting less catastrophic forgetting and reaching results comparable to models trained on a static training set which consisted of data from all domains. Similar effects occurred in lung nodule detection in CT, and results showed that using pseudo-domains (DM-PD) led to fewer false-positive detections, than standard dynamic memory.

Our approach is a step toward the design of a strategy to learn on a continuous data stream of medical images that can potentially be deployed in clinical practice. Nevertheless, the method has several limitations. First, more research is needed to demonstrate that we can design systems that guarantee that there is no catastrophic forgetting in the future when the number of scanner is scaled up substantially. Proving that the performance of DL models will not decline with future domains is challenging. Second, DM requires storing a subset of the images for rehearsal during training. While this rehearsal set is substantially smaller than the entire data set, privacy concerns or storage limitations may become relevant. Finally, we do not take the cost of annotating cases into account, assuming that there are target labels or bounding boxes for training available for each sample in the data stream. In clinical practice, this assumption does not hold, and a human-in-the-loop concept[7] such as active learning[46] is needed to collect new annotations for unknown domains economically.

## Methods

**Dynamic memory.** The goal of our method[33] is to continuously update the parameters $\theta$ of a task model, already trained on a base training set of one domain, on a continuous data stream with multiple but unknown imaging domains. Training data to update the $\theta$ is composed to capture novel data characteristics while sustaining the diversity of the overall training corpus. At each step, examples from previously seen data sampled from memory $\mathcal{M}$ and new examples (input-mini-batch $\mathcal{B}$) form the training data (training-mini-batch $\mathcal{T}$) for updating the $\theta$. The dynamic memory (DM) $\mathcal{M} = \{\langle \mathbf{m}_1, n_1 \rangle, \dots, \langle \mathbf{m}_M, n_M \rangle\}$ is holding image-target pairs $\langle \mathbf{m}, n \rangle$ of a fixed-size $M$ that are stored and updated during continual training. The DM approach is used to keep $\mathcal{M}$ diverse and representative of the visual variations across all domains. The important step in this procedure is to decide which image-target pairs to keep in memory, without explicit domain knowledge. This procedure is depicted in Fig. 7. We apply two simple rules to update $\mathcal{M}$: (1) every novel image-target pair is stored to $\mathcal{M}$. (2) the image replaced in memory is close to the novel image according to a high-level style metric. The high-level metric in rule 2 is critical, as it ensures that the memory is representative and diverse. To evaluate the style of an image, following neural style transfer as proposed in ref. [47], we define a metric based on the Gram matrix $G^l \in \mathbb{R}^{N_l \times N_l}$ where $N_l$ is the number of feature maps in layer $l$ of a pre-trained style model. This style model is pre-trained on ImageNet and its weights remain fixed during continual training. Given an input image $\mathbf{x}$, $G^l_{ij}(\mathbf{x})$ is defined as the inner product between the vectorized activations $\mathbf{f}_{il}(\mathbf{x})$ and $\mathbf{f}_{jl}(\mathbf{x})$ of two feature maps $i$ and $j$ in a layer $l$:

$$G^l_{ij}(\mathbf{x}) = \frac{1}{N_l M_l} \mathbf{f}_{il}(\mathbf{x})^\top \mathbf{f}_{jl}(\mathbf{x}), \quad (1)$$

where $M_l$ denotes the number of elements in the vectorized feature map (width × height). With the Gram matrix, we define a Gram distance $\delta(\mathbf{x}, \mathbf{y})$ between two images $\mathbf{x}$ and $\mathbf{y}$ for a set of convolutional layers $\mathcal{L}$ as:

$$\delta(\mathbf{x}, \mathbf{y}) = \sum_{l \in \mathcal{L}} \frac{1}{N_l^2} \sum_{i=1}^{N_l} \sum_{j=1}^{N_l} (G^l_{ij}(\mathbf{x}) - G^l_{ij}(\mathbf{y}))^2. \quad (2)$$

During continual training, at each step an input-mini-batch $\mathcal{B} = \{\langle \mathbf{b}_1, c_1 \rangle, \dots, \langle \mathbf{b}_B, c_B \rangle\}$ of $B$ cases (image $\mathbf{b}$ and target $c$) is taken from the data stream. Sequentially, each element of $\mathcal{B}$ replaces the (according to Eq. (2)) closest

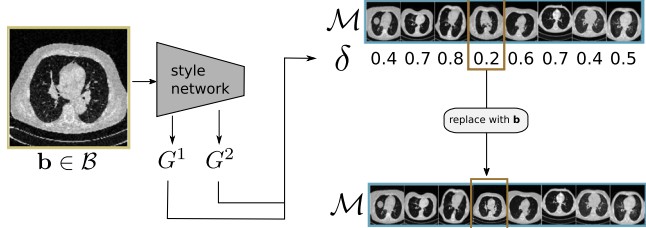

**Fig. 7 Updating dynamic memory $\mathcal{M}$ of size 8 with a new sample b, taken from the input-mini-batch.** (1) The image is given to the style model and the Gram matrices $G^1$, $G^2$ are calculated. (2) Based on those matrices the Gram distance $\delta$ between **b** and each element in the memory is calculated. (3) **b** replaces the memory element with minimal distance $\delta$.

element of $\mathcal{M}$. Formally: given an input image-target pair $\langle \mathbf{b}_i, c_i \rangle$, the sample will replace the element in $\mathcal{M}$ with index

$$\xi(i) = \arg\min_j \delta(\mathbf{b}_i, \mathbf{m}_j) \mid j \in \{1, \dots, M\}. \tag{3}$$

This replacement strategy is applied after an initial phase of continual training in which the memory is filled with elements of the data stream. After the memory is updated, a training-mini-batch $\mathcal{T} = \{\langle \mathbf{t}_1, u_1 \rangle, \dots, \langle \mathbf{t}_T, u_T \rangle\}$ of size $T$ is assembled. Each element of $\mathcal{B}$ for which the current task model performs poorly according to the task metric is added to $\mathcal{T}$ and additional cases are randomly drawn from $\mathcal{M}$ until $|\mathcal{T}| = T$. Finally, the parameters of the task model are updated using the training-mini-batch $\mathcal{T}$ to perform forward and backward pass.

*Pseudo-domain module.* As an optional element of the method, we develop a pseudo-domain module that identifies pseudo-domains, serving as a proxy for the unknown, real-world domains. Those pseudo-domains are used to balance the memory $\mathcal{M}$ and $\mathcal{T}$ during continual training.

We define the set of pseudo-domains as $\mathcal{D} = \{\mathbf{i}_1 \dots \mathbf{i}_D\}$. Where $\mathbf{i}_j$ is a trained Isolation Forest (IF)[48] used as one-class anomaly detection for the pseudo-domain $j \in \{1, \dots, D\}$. We use IFs, because of their simplicity and the good performance on small sample sizes. In order to use IF a dimensionality reduction of the Gram matrix is needed, we define a Gram embedding $e(\mathbf{x})$ as a reduced version of the Gram matrix (Eq. (1)) using Sparse Random Projection (SRP)[49] fitted to the base training set. An image $\mathbf{x}$ is assigned the pseudo-domain maximizing the decision function of $\mathbf{i_d}$:

$$p(\mathbf{x}) = \begin{cases} \arg\max_d \mathbf{i_d}(e(\mathbf{x})) & \text{if } \max_d [\mathbf{i_d}(e(\mathbf{x}))] > 0 \mid d \in \{1, \dots, D\} \\ -1 & \text{otherwise.} \end{cases} \tag{4}$$

If $p(\mathbf{x}) = -1$ the image-target pair is added to the outlier memory $\mathcal{O}$. Within $\mathcal{O}$ we identify new pseudo-domains to add to $\mathcal{D}$.

*Discovery of pseudo-domains in $\mathcal{O}$*: $\mathcal{O}$ holds training pairs that do not fit an already identified domain and might form a new domain. Examples are stored until they are assigned a new pseudo-domain or if a fixed number of training steps is reached. If no new pseudo-domain is discovered for an image it is considered a 'real' outlier and removed from the outlier memory. The discovery process is started when $|\mathcal{O}| = o$, where $o$ is a fixed threshold. A check if a dense region is present in the memory is done by calculating the pairwise Euclidean distances of all elements in $\mathcal{O}$. If there is a group of images where the distances are below a threshold $t$ a new IF $\mathbf{i_n}$ is fitted to the Gram embeddings of the dense region and the set of pseudo-domains $\mathcal{D}$ is updated. Samples belonging to the new pseudo-domain are transferred from $\mathcal{O}$ to $\mathcal{M}$, balancing the memory $\mathcal{M}$ such that each domain $d \in \mathcal{D}$ occupies at least $\frac{M}{|\mathcal{D}|}$ positions.

*Memory update with pseudo-domains*: To use the pseudo-domain module to balance training we define $\mathcal{M}_d = \{\langle \mathbf{m}, n \rangle \in \mathcal{M} | p(\mathbf{m}) = d\}$ as the subset of $\mathcal{M}$ where the pseudo-domain is $d$. And extend the rules to update $\mathcal{M}$ outlined in the previous section as follows, a new image-target pair $\langle \mathbf{x}, y \rangle$ is inserted into memory: If $|\mathcal{M}_d| < \frac{M}{|\mathcal{D}|}$ where $d = p(\mathbf{x})$ we replace a random element with a different pseudo-domain for which $|\mathcal{M}_r| > \frac{M}{|\mathcal{D}|}$. Otherwise, we replace an element according to Eq. (3) using only $\mathcal{M}_d$ instead of the whole memory $\mathcal{M}$.

**Experimental setup.** All networks used are implemented in Python 3.6 with PyTorch 1.6.0[50] using the implementation within the torchvision package. As a style network, we utilize ResNet-50[51] pre-trained on ImageNet for all experiments. For cardiac segmentation, the task network is a fully-convolutional network model with a ResNet-50 backbone[21]. For lung nodule detection we use a Faster-RNN with ResNet-50 backbone[22].

**Evaluation.** We evaluate the ability of continual learning to improve performance on previously seen domains by adding new domains backward transfer (BWT), and the contribution of previous domains in the training data to improving the

accuracy on subsequent domains forward transfer (FWT) following the definitions in ref. [19]. BWT measure how learning a new domain influences the performance on previous tasks, FWT quantifies the influence on future tasks. Negative BWT values indicate catastrophic forgetting, thus avoiding negative BWT is especially important for continual learning.

*Dice coefficient.* For cardiac segmentation Dice coefficient or Dice score (DSC) is used as a performance metric. The DSC measures the overlap as:

$$D(X, Y) = \frac{2|X \bigcap Y|}{|X| + |Y|}, \tag{5}$$

where $X$ and $Y$ are segmentation maps, such as predicted and ground-truth cardiac segmentation.

*Average precision.* For lung nodule detection we use average precision (AP) as the performance metric, to judge the performance of the models with a single metric. We follow the AP definition in ref. [52]. The intersect over union between boxes of ground truth and prediction has to be over 0.3 to be counted as a true positive, otherwise, the prediction will be regarded as a false positive. The precision is then averaged at eleven equally spaced recall levels:

$$\text{AP} = \frac{1}{11} \sum_{r \in \{0, 0.1, \dots, 1\}} p_i(r), \tag{6}$$

where $p_i$ is defined as:

$$p_i(r) = \max_{r*: r* \geq r} p(r*), \tag{7}$$

the maximum precision for which the corresponding recall exceeds the threshold.

**Reporting summary.** Further information on research design is available in the Nature Research Reporting Summary linked to this article.

## Data availability

The cardiac MRI data that support the findings of this work are part of the Multi-Centre, Multi-Vendor & Multi-Disease Cardiac Image Segmentation Challenge (M&Ms)[14] and are publicly available (https://www.ub.edu/mnms/). The CT data that support the findings is provided as Lung Image Database Consortium image collection (LIDC-IDRI) as part of the cancer imaging archive and are publicly available (https://wiki.cancerimagingarchive.net/display/Public/LIDC-IDRI); and as LNDb Challenge[17] data (https://lndb.grand-challenge.org/).

## Code availability

The custom code that supports the findings of this study is publicly available at Github (https://github.com/cirmuw/dynamicmemory)[53].

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

## Acknowledgements

This study has received partial funding from the European Commission (Bigmedilytics 780495, TRABIT 765148), the Austrian Science Fund (FWF) (P 35189, P 34198), Austrian National Bank Anniversary Fund (18207), Vienna Science and Technology Fund (WWTF) Project Nr. LS20-065 (PREDICTOME). Furthermore, this study was partially supported by the Novartis Pharmaceuticals Corporation (M.P., G.L.). This study was partially supported by Boehringer Ingelheim RCV GmbH & Co KG (BI) (H.P.). BI had no role in the design, analysis or interpretation of the results in this study. BI was given the opportunity to review the manuscript for medical and scientific accuracy. Part of the computations for research was performed on GPUs donated by NVIDIA.

## Author contributions

M.P., J.H. and G.L. contributed to the conception and design of the study. M.P. and J.H. developed the deep learning model. M.P. performed the experiments. M.P., J.H. and G.L. reviewed the experiment results. All the authors provided advice and contributed in preparing the manuscript.

## Competing interests

Authors declare the following competing interests: M.P., J.H. and O.P. declare no competing interests. C.H.: Research Consultant for Siemens Healthineers and Bayer Healthcare, Stock holder at Hologic Inc. J.A.B.: Member of the Board of Directors for Accumen, Inc. H.P.: Speakers Honoraria for Boehringer Ingelheim and Roche. Received a research grant from Boehringer Ingelheim. G.L.: Co-founder and stock holder at contextflow GmbH. Received research funding from Novartis Pharmaceuticals Corporation.
