## [Peer Review File · Nature Communications]

Reviewers' Comments:

Reviewer #1:

Remarks to the Author:

The authors present a rehearsal based continual learning method for dealing with continually shifting domains in medical imaging.

They incorporate a dynamic memory which stores small but diverse set of examples to avoid catastrophic forgetting.

An important component of the method is a style metric that identifies variety of styles in the data stream. Here changes in style could be recognized as changes in the domain.

A weakness of the method is that while the method seems to be robust in style shift, it is not robust to concept shift (eg, change in nodule size)

The paper is good , the experiments are sound and thorough. One potential experiment that is missing is the ablation study on the style metric since its such an important component in the paper. There are other rehearsal methods such as Lopez-Paz et al. Or even use a random subset of examples instead of using the syle metric to select them.

Lopez-Paz, D. and Ranzato, M.A., 2017. Gradient episodic memory for continual learning. arXiv preprint arXiv:1706.08840.

Reviewer #2:

Remarks to the Author:

The author proposed a continual learning-based framework for the domain shift (data from various vendors/scanners) problems in medical image analysis tasks, including novel dynamic memory and pseudo-domain modules to preserve the representative and effective data for continuous training when new domain data are inputted for the training. The authors applied and experimented with the proposed method in two applications, i.e., cardiac segmentation in MRs and lung nodule detection in CTs. Two public datasets are employed here for the experiments, both of which contain data from different vendors. Superior results have been reported in comparison to a naive continual learning baseline (that do not contain the proposed memory and pseudo-domain modules). Ablation studies are also conducted to prove the effectiveness of each part, including different memory sizes (64/128 entries) and with/without pseudo-domain module. The manuscript is overall well-written and organized. The introduction is quite clear and easy to follow. However, several things need to be further addressed before publication. Details are listed as follows:

1. The memory is indeed a selected set of all training data (from different vendors). The authors experimented with two different sizes (64 and 128). At the same time, I am curious how large the memory is needed to get the closest results to the model trained with all data (JModel as presented)? is it different for various datasets?
2. In comparison to other domain adaptation methods, the nature of continual learning actually limits the reuse of trained model for learning a new model on a different site/location, where the original data (memory) and learned pseudo domain are not available. I hope the author could comment and discuss this limitation.
3. How will the amount of data from each vendor affect the learning, especially for the base model. Situations may vary greatly in practice, which need to be experimented with and discussed further, e.g., the base model is trained with a much larger amount of data compared to follow-up vendors, or the data amount of one of the follow-up vendors is much larger than the others.
4. The experimental comparison of the proposed methods to other previous ones (SOTAs) needs to be enhanced, e.g., only one method (EWC) is compared for the cardiac segmentation application, and none for the lung nodule detection task. Other learning without forgetting methods (e.g.,

[23]) or general domain adaptation methods need to be also evaluated here. Otherwise, it is hard to appreciate the advantage of the proposed method, considering extra info (memory and pseudo domain) is required to retrain the model with other sites/domains data.

Reviewer #3:

Remarks to the Author:

This paper proposes a continual learning approach with dynamic memory to alleviate catastrophic forgetting for medical imaging tasks. By doing so, the models are adapted to emerging variations in a continuous data stream while counteracting catastrophic forgetting. Experiments are conducted on two popular medical imaging tasks, i.e. segmentation and detection.

Overall, the idea is interesting, the proposed method is novel and effective, and the presentation is generally clear. The discussions are helpful. I have the following concerns that might help to improve the quality:

1. The motivation is not clearly explained. It would be much better to quantitatively demonstrate that continual learning is necessary and important. For example, the results of direct transfer and continual learning are compared in the Introduction section. Some detailed segmentation results would also help.

2. The compared baselines seem to be out-of-date. EWC baseline is published in 2017. More recent baselines are recommended to better demonstrate the effectiveness of the proposed method.

3. It would be better to put the datasets in the experimental setting section instead of the method section.

4. When showing the discussions, some closely related methods are missing, such as "Domain Adaptation for Medical Image Analysis: A Survey", "A Review of Single-Source Deep Unsupervised Visual Domain Adaptation" on domain adaptation.

5. The presentation still needs proofreading. For example, some equations are punctuated and some are not. In figure 3, the legend colors do not match with the colors in the main figure.

Point by point answers to the reviewers

Reviewer #1 (Remarks to the Author):

The authors present a rehearsal based continual learning method for dealing with continually shifting domains in medical imaging. They incorporate a dynamic memory which stores small but diverse set of examples to avoid catastrophic forgetting. An important component of the method is a style metric that identifies variety of styles in the data stream. Here changes in style could be recognized as changes in the domain. A weakness of the method is that while the method seems to be robust in style shift, it is not robust to concept shift (eg, change in nodule size)

REV1-1: *The paper is good , the experiments are sound and thorough. One potential experiment that is missing is the ablation study on the style metric since its such an important component in the paper. There are other rehearsal methods such as Lopez-Paz et al. Or even use a random subset of examples instead of using the style metric to select them.*

Lopez-Paz, D. and Ranzato, M.A., 2017. Gradient episodic memory for continual learning. arXiv preprint arXiv:1706.08840.

Thank you for your suggestion, we agree that using a random replacement strategy is useful to evaluate the effect of the proposed style metric. We added an ablation experiment in Section 2.2. and Section 2.3. in which we replaced the samples in memory at random (random replacement). This showed that our style metric is effective in diversifying the memory in a way that is beneficial for continual training.

We added results for Gradient episodic memory (GEM) as an additional reference method. In contrast to DM/DM-PD, GEM requires domain membership knowledge for the training of models, yet DM/DM-PD performs competitively.

Reviewer #2 (Remarks to the Author):

The author proposed a continual learning-based framework for the domain shift (data from various vendors/scanners) problems in medical image analysis tasks, including novel dynamic memory and pseudo-domain modules to preserve the representative and effective data for continuous training when new domain data are inputted for the training. The authors applied and experimented with the proposed method in two applications, i.e., cardiac segmentation in MRs and lung nodule detection in CTs. Two public datasets are employed here for the experiments, both of which contain data from different vendors. Superior results have been reported in comparison to a naive continual learning baseline (that do not contain the proposed memory and pseudo-domain modules). Ablation studies are also conducted to prove the effectiveness of each part, including different memory sizes (64/128 entries) and with/without pseudo-domain module. The manuscript is overall well-written and organized. The introduction is quite clear and easy to follow. However, several things need to be further addressed before publication. Details are listed as follows:

REV2-1: *The memory is indeed a selected set of all training data (from different vendors). The authors experimented with two different sizes (64 and 128). At the same time, I am curious how*

large the memory is needed to get the closest results to the model trained with all data (JModel as presented)? is it different for various datasets?

This is a relevant point, and in the revised manuscript we have added experimental results for different memory sizes to evaluate the impact of memory size (Suppl. Table S4, Table S5). The results demonstrate that in general, the larger the memory size, the better the model performance. Note that due to differences in the training sequence, even when storing all samples to memory, the continual learning approach will not match the JModel.

To clarify this, for lung nodule detection experiments we have added the following explanation in Section 2.3:

“Supplemental table S5 shows results of comparing different memory sizes $M=\{64, 128, 256, 512, 883\}$ (883 corresponds to storing all training samples to memory). For DM, a larger memory size was beneficial compared to a smaller memory size. For DM-PD, the results showed that the smaller the memory ($M=64$ and $M=128$), the more performance gain was achieved by pseudo-domain (PD) detection compared to mere DM. For larger memory sizes, the benefit of PDs vanished. Due to differences in training sequence, the performance of DM with $M=883$ is different from those of JModel. While DM/883 performed better on Scanner E and F, JModel showed higher AP values for Scanner G and H.”

REV2-2: In comparison to other domain adaptation methods, the nature of continual learning actually limits the reuse of trained model for learning a new model on a different site/location, where the original data (memory) and learned pseudo domain are not available. I hope the author could comment and discuss this limitation.

This is correct, our continual learning approach assumes continued availability of a (reduced) set of cases. If models have to be transferred to a new site, and access to prior training data is cut-off, transfer learning is a suitable approach. Here, only the model itself is kept.

Having said that, since during training we expose the model to diverse data from a variety of different scanners, we speculate that the model weights (without memory) are well suitable for transfer to a different site.

We have clarified this in the discussion (Section 3) of the manuscript:

“A third related area is transfer learning [51]. Here, an existing model is transferred to a new task or domain by fine-tuning on new data. The sole aim is a model performing well there, regardless of its capability to work well on the initial domain. During fine-tuning no data from the initial domain is necessary.”

REV2-3: How will the amount of data from each vendor affect the learning, especially for the base model. Situations may vary greatly in practice, which need to be experimented with and discussed further, e.g., the base model is trained with a much larger amount of data compared to follow-up vendors, or the data amount of one of the follow-up vendors is much larger than the others.

Thank you for pointing this out. The amount of data from each vendor has an influence on the training. The base model was trained in an epoch-based training procedure until convergence on a

larger data set, to build an initial base model before exposing it to a continuous stream of data in clinical practice. Here, we focussed on strategies to adapt to new data, and thus wanted the initial model to be as optimal as the available data allowed for.

Impact of sample sizes during the continual data stream is a relevant point. In our experiments in Section 2.2. and Section 2.3. we showed that our method counteracted the influence of differences in sample sizes between vendors by detecting the style of images (and subsequently pseudo-domains) and keeping a diverse, balanced memory over all different styles. This is further enhanced in the pseudo-domain approach.

To clarify the influence of the amount of different follow-up vendors based on the experimental results, we have added two paragraphs to the result section: For cardiac segmentation in Section 2.2. we have added:

“On data from Scanner B, for which a relatively small sample size of 720 images was used for training, DM and DM-PD were able to achieve good performance without domain membership information. Learning with random replacement, EWC and naive training resulted in a significant performance drop for Scanner B. This demonstrated that by using a Gram matrix based style metric, DM was less sensitive to the amount of samples per scanner vendor than other continual learning strategies.”

For lung nodule detection in Section 2.3. the following explanation has been appended:

“The random replacement strategy struggled with learning domains that were present less frequently in the training (Scanner F and G) as they were replaced over time by Scanner H data, resulting in forgetting on those scanners. This effect was less severe for Scanner E data since the base training was performed on data of this scanner. DM and DM-PD counteracted this forgetting by using a style based metric to diversify the memory and thus kept samples of all scanners in memory.”

REV2-4: *The experimental comparison of the proposed methods to other previous ones (SOTAs) needs to be enhanced, e.g., only one method (EWC) is compared for the cardiac segmentation application, and none for the lung nodule detection task. Other learning without forgetting methods (e.g., [23]) or general domain adaptation methods need to be also evaluated here. Otherwise, it is hard to appreciate the advantage of the proposed method, considering extra info (memory and pseudo domain) is required to retrain the model with other sites/domains data.*

We have expanded the experiment section, including two additional continual learning methods, gradient episodic memory (GEM) and experience replay with maximally inferred retrieval (ER-MIR) to our evaluation in Sections 2.2 (Table 2) and 2.3 (Table 3). We have clarified in the manuscript that they need domain membership information during training, which is not the case for the two proposed approaches.

Reviewer #3 (Remarks to the Author):

This paper proposes a continual learning approach with dynamic memory to alleviate catastrophic forgetting for medical imaging tasks. By doing so, the models are adapted to emerging variations in a continuous data stream while counteracting catastrophic forgetting. Experiments are conducted on two popular medical imaging tasks, i.e. segmentation and detection. Overall, the idea is

interesting, the proposed method is novel and effective, and the presentative is generally clear. The discussions are helpful. I have the following concerns that might help to improve the quality:

REV3-1: *The motivation is not clearly expalined. It would be much better to quantitatively demonstrate that continual learning is necessary and important. For example, the results of direct transfer and continual learning are compared in the Introduction section. Some detailed sgementation results would also help.*

We have added results comparing the continual learning approaches, with a base model only trained on the initial training data. The key point is that the proposed approach adapts to new scanner types, while at the same time not *forgetting* to work well on scanners seen during the initial phase of training. Tables 2 and 3 now contain results comparing the proposed approaches (DM and DM-PD) with alternative continual learning approaches and with a static base model. We now also refer to these results from the introduction.

REV3-2: *The comapred baseliens seem to be out-of-date. EWC baseline is published in 2017. More recent baseliens are recommended to better demonstrate the effectiveness of the proposed method.*

For better comparison we added gradient episodic memory (GEM) and experience replay with maximally inferred retrieval (ER-MIR) to our evaluation in Sections 2.2. and 2.3. Both require domain membership information during training. We have updated the results and discussion accordingly.

REV3-3: *It would be better to put the datasets in the experimental setting section instead of the method section.*

Thank you for your suggestion, we have adapted the manuscript accordingly. The dataset description is now Section 2.1..

REV3-4: *When showing the discussions, some closely related method are misssing, such as "Domain Adaptation for Medical Image Analysis: A Survey", "A Review of Single-Source Deep Unsupervised Visual Domain Adaptation" on domain adaptation.*

Thank you for this suggestion. We have added the reference to the manuscript and have extended the explanation regarding the differences of domain adaptation and our continual learning setting in the discussion (Section 3) :

“Domain adaptation (DA) is a related area of research dealing with domain shifts [8, 48, 49]. DA focuses on adapting the knowledge learned on one or multiple source domains to a target domain. In medical imaging DA is used to adapt between different imaging modalities or different image acquisition settings [16].”

REV3-5: *The presentation still needs proofreading.-For example, some equations are punctuated and some are not. In fugire 3, the legend colors are not match with the colors in the main figure.*

Thank you for pointing out those mistakes. We performed proofreading on the revised manuscript, and corrected the mistakes mentioned in the comment.

POLICIES AND FORMS REQUIRED FOR RESUBMISSION

We have included a Data Availability and a Code Availability section in the manuscript.
The code for the paper is available here: <https://github.com/cirmuw/dynamicmemory>

The results are obtained with experiments performed on publicly available data. The cardiac MRI data is part of the Multi-Centre, Multi-Vendor & Multi-Disease Cardiac Image Segmentation Challenge (M&Ms) and is publicly available (<https://www.ub.edu/mnms/>). The CT data that support the findings is provided as Lung Image Database Consortium image collection (LIDC-IDRI) as part of the cancer imaging archive and are publicly available (<https://wiki.cancerimagingarchive.net/display/Public/LIDC-IDRI>) and as LNDb Challenge data (<https://lndb.grand-challenge.org/>).

All plots have been adjusted according to the policies, including plotting individual data points for small sample sizes.

Reviewers' Comments:

Reviewer #2:

Remarks to the Author:

To address the reviewer's concerns, the authors did a major revision (i.e., additional experiments about memory size, vendor data imbalance, and more comparison/discussion to previous methods). Thus, I believe the manuscript is overall in good shape now and ready for publication.

Just one minor point is listed below:

Why do the results for "base" not have the STDs in Table 2/3, any particular reasons?

Reviewer #3:

Remarks to the Author:

In the revised version, the authors have addressed my previous concerns and therefore I recommend acceptance. In the final version, the format of references should be made consistent. For example, the journal or conference names of some references [50] are missing.

Point by point answers to the reviewers

Reviewer #2 (Remarks to the Author):

To address the reviewer's concerns, the authors did a major revision (i.e., additional experiments about memory size, vendor data imbalance, and more comparison/discussion to previous methods). Thus, I believe the manuscript is overall in good shape now and ready for publication. Just one minor point is listed below:

REV2-1: Why do the results for "base" not have the STDs in Table 2/3, any particular reasons?

For base training we do not have STDs in Table 2 and 3, as we run base training only once for each task and use this base network for all subsequent experiments. This ensures a fair comparison between continual learning methods, as the results of base training do not influence the final outcome differently.

We added the following sentence in the caption of Table 2 and 3:

“For base training only one model was trained to avoid influence of base training results on subsequent continual training, therefore no standard deviations are indicated.”

Reviewer #3 (Remarks to the Author):

In the revised version, the authors have addressed my previous concerns and therefore I recommend acceptance.

REV3-1: In the final version, the format of references should be made consistent. For example, the journal or conference names of some references [50] are missing.

Thank you for pointing this out. We updated the reference list accordingly to make sure the format of references is consistent.